# Cross-cultural adaptation of the 5-Question Stigma Indicators in trachoma-affected communities, Ethiopia

**Misrak Negash[1], Zerihun Tadesse[2], Fentie Ambaw[3], Michael Beka[4], Tilahun Belete[4], Melkamu Abte[5], Kebede Deribe[6], Julian Eaton[7,8], Eve Byrd[9], E. Kelly Callahan[10], David Addiss[11], Wim H. van Brakel[12], Abebaw Fekadu[13], David Macleod[14], Matthew Burton[14], Esmael Habtamu[14,15,16]***

**1** Department of Psychiatry, College of Medicine and Health Science, Dilla University, Dilla, Ethiopia, **2** The Carter Center, Addis Ababa, Ethiopia, **3** School of Public Health, College of Medicine and Health Science, Bahir Dar University, Bahir Dar, Ethiopia, **4** Department of Psychiatry, College of Medicine and Health Science, Bahir Dar University, Bahir Dar, Ethiopia, **5** Amhara Regional Health Bureau, Bahir Dar, Ethiopia, **6** Children's Investment Fund Foundation, Addis Ababa, Ethiopia, **7** Centre of Global Mental Health, London School of Hygiene & Tropical Medicine, London, United Kingdom, **8** CBM Global, Cambridge, United Kingdom, **9** Mental Health Program, The Carter Center, Atlanta, Georgia, United States of America, **10** Trachoma Control Program, The Carter Center, Atlanta, Georgia, United States of America, **11** Focus Area for Compassion and Ethics (FACE), The Task Force for Global Health, Decatur, Georgia, United States of America, **12** NLR, Amsterdam, The Netherlands, **13** Department of Psychiatry, Addis Ababa University, Addis Ababa, Ethiopia, **14** International Centre for Eye Health, London School of Hygiene & Tropical Medicine, London, United Kingdom, **15** Department of Ophthalmology, College of Medicine and Health Science, Bahir Dar University, Bahir Dar, Ethiopia, **16** Eyu-Ethiopia, Bahir Dar, Ethiopia

* esmael.ali@lshtm.ac.uk

**Data Availability Statement:** Data is available to any researcher under reasonable request. The Amhara Regional Health Bureau Ethics Committee requires that all data sharing requests are reviewed

## Abstract

Stigma is common in people affected with Neglected Tropical Diseases (NTDs). However, no validated tools are available to assess and monitor stigma in trachoma-affected communities. We tested the cross-cultural equivalence of the 5-question stigma indicator-affected persons (5-QSI-AP) scale in persons with trachomatous trichiasis (TT), the blinding stage of trachoma, and the 5-question stigma indicator-community stigma (5-QSI-CS) scale in person without TT, in Amhara region, Ethiopia. Conceptual, item, semantic, and operational equivalence were assessed through exploratory qualitative methods; measurement equivalence was assessed quantitatively through internal consistency, construct validity, and reproducibility. A total of 390 people participated: 181 were persons with TT, 182 persons without TT, 19 mental health, trachoma, social science, and linguistics experts, and eight interviewers. Items included in both scales were adequately relevant and important to explore stigma in the target culture. Concern about others knowing that they have TT, shame, avoidance by others, and problems getting married or in their marriage were among the issues persons with TT faced in this study community. The 5-QSI-AP had a Cronbach's α of 0.57 for internal consistency and showed adequate discriminant validity where persons with central corneal opacity from TT had higher mean stigma scores than their counterparts. The 5-QSI-CS had a Cronbach's α of 0.70 for internal consistency and a correlation of r = 0.23 with the Social Distance Scale (SDS) for convergent validity. The test-retest reliability analysis between the initial and repeat measures produced an intraclass correlation

and approved by them before data can be shared. To facilitate the data access process please contact ethics@lshtm.ac.uk.

**Funding:** This work received financial support from the Coalition for Operational Research on Neglected Tropical Diseases (reference # NTDSC 194D to EH and ZT), which is funded at The Task Force for Global Health primarily by the Bill & Melinda Gates Foundation, by the United States Agency for International Development through its Neglected Tropical Diseases Program, and with UK aid from the UK Government. EH is a Wellcome International Intermediate Fellowship Fellow [Grant Number 221991/Z/20/Z], jointly funded by Wellcome and the Department of Health and Social Care (DHSC), through the National Institute for Health Research (NIHR) (using the UK's Official Development Assistance (ODA) Funding). The funders had no role in study design, data collection and analysis, decision to publish, or preparation of the manuscript. The findings and conclusions contained within are those of the authors and do not necessarily reflect positions or policies of the funders.

**Competing interests:** The authors have declared that no competing interests exist.

coefficient of 0.60 and 0.53 for the 5-QSI-AP and 5-QSI-CS respectively, and no evidence of systematic bias in mean stigma scores. The 5-QSI scales have satisfactory cultural validity to assess and monitor stigma in this trachoma-affected Amharic-speaking study population. With further cross-cultural validation, these brief and easy to administer scales would offer the possibility to rapidly measure and monitor stigma associated with NTDs.

## Introduction

Trachoma is one of the most common Neglected Tropical Diseases (NTDs) and the leading infectious cause of blindness worldwide [1]. The blinding stage of trachoma, trachomatous trichiasis (TT) is characterized by in-turning eyelashes constantly scratching the eyeball, causing painful corneal abrasions and ulcerations, which eventually result in corneal opacity leading to irreversible visual impairment [2]. Women are disproportionately more affected than men, up to three times higher in settings like Ethiopia [3]. TT causes considerable pain which can have major physical, social, psychological, and economic consequences for affected families and communities even prior to the development of visual impairment [4]. Various studies reported that TT causes functional and physical impairment, an inability to work and earn an income from the chronic pain and vision impairment, and social withdrawal and exclusion from internalized stigma [5–7]. A recent case-control study reported that TT adversely impacts vision and health-related quality of life (QoL), even before visual impairment develops [4].

Persons with TT can be stigmatized in relation to the unsightly photophobic, watery, and purulent appearance of their eyes, other community members' fear of contracting the disease, inability to fulfill certain gender roles, and being a social and financial burden to their families due to limited engagement in productive activities [5]. A study conducted in Niger revealed that persons with TT are stigmatized by members of their community, manifested by their group of friends 'talking about them, how dirty they are and spilling their tears on everything they touch' and having difficulties in getting married or in an existing marriage [5]. There was also a report of internalized stigma which resulted in social withdrawal and limiting oneself from participating in village activities in response to anticipated stigmatization from their community [5]. Such stigmatization may sustain suffering through feelings of shame, low self-esteem, and emotional distress leading to mental health disorders and disability. However, despite these possible associations between trachoma and stigma, neither are culturally validated quantitative tools available nor have adequate attempts been made to quantify and monitor stigma in persons affected with TT and the communities where they live. The purpose of this study is to fill this gap by validating a quantitative stigma tool that can help to rapidly measure and monitor stigma associated with TT and the impact that the surgical service provided to treat TT has on stigma. The data generated using adequately validated tools would be used to encourage allocation of resources to address the unmet mental healthcare needs of trachoma and other NTDs affected communities and thereby facilitate and support the elimination of these diseases by 2030.

There are several stigma measurement tools. The most used in NTD affected communities are Stigma Assessment and Reduction of Impact (SARI, 21 item scale), Explanatory Model Interview Catalogue (EMIC, 15 item scale), the Social Distance Scale (SDS, seven item), and the 5-Question Stigma Indicators (5-QSI, 5 item) [8]. The 5-QSI not only have the fewest items but also the easiest to administer. The 5-QSI scales are mentioned in the 'suggested actions' section of the Global Leprosy Strategy 2016–2020 Monitoring & Evaluation Manual as an easy-to-use questionnaire for stigma assessment [9]. The relative ease of administration

makes the 5-QSI ideal tools to measure and monitor stigma in NTDs, raising the need to expand validation in diverse communities and health conditions, which is why it is selected for cross-cultural adaption in this study.

The (5-QSI) were developed by van Brakel at the request of the World Health Organization (WHO) Global Leprosy Programme in 2016 based on the Explanatory Model Interview Catalogue (EMIC) scales. They were published in 2017 as part of the WHO Monitoring and Evaluation Guide accompanying the WHO Global Leprosy Strategy 2016–2020 [9]. The scales were validated in Nepal and India to assess and monitor leprosy-related stigma in both the community and affected persons [10–12]. Here we report the results of the cross-cultural adaptation of the English version of the 5-QSI into the Amharic language for use to measure and monitor trachoma related stigma in the most historically trachoma-affected setting, Amhara Region, Ethiopia.

## Materials and methods

### Study design

A community-based mixed-method cultural equivalence study was employed to validate the 5-QSI scales in trachoma-affected communities using the Universalist Validation Framework developed by Herdman et al. [13, 14]. A qualitative exploratory study was employed to determine conceptual, item, semantic, and operational equivalence, while a quantitative study was employed to determine measurement equivalence of the 5-QSI scales.

### Study site

The study was conducted in South Mecha, Bahir Dar Zuria, and North Achefer districts of West Gojjam zone, Amhara Region, Ethiopia. As of 2020, West Gojjam has a total population of about 2.8 million with ~83% rural inhabitants (S1 Data). The largest ethnic group is the Amhara, with Amharic spoken as a first language by more than 90% of the inhabitants. West Gojjam zone is historically one of the most trachoma-affected zones of Amhara Regional State. A population-based trachoma impact survey conducted between 2010 and 2015 reported a 2.9% prevalence of TT in persons ages 15 years and above [15].

### Study participants

Study participants included: persons with TT; persons without TT; and experts in mental health, trachoma, social science, and linguistics who participated in the translation, review, and panel discussion of the stigma tools; and research data collectors to test operational equivalence.

Persons with TT were identified through active community-based screenings in high-burden districts in non-probability sampling until the required sample size was reached. Inclusion criteria for persons with TT included those (i) adults ≥18 years of age, (ii) with un-operated vision-threatening trichiasis defined as (a) ≥1lash touching the cornea and with moderate to severe entropion in the upper eyelid, or (b) with evidence of epilation (the repeated plucking, cutting or burning of eyelashes touching the eye) on ≥1/3 of the upper eyelid and with moderate to severe entropion, or (c) ≥6 peripheral lashes, (iii) with vision-threatening postoperative TT defined as (a) ≥5 eyelashes arising from the upper eyelid touching the eye among which ≥2 entropic lashes are touching the cornea, or (b) with evidence of epilation on ≥1/3 of the eyelid, or (c) ≥6 peripheral postoperative lashes, (iv) consented to participate in the study, (v) able to speak Amharic language Exclusion criteria for persons with postoperative TT included those with two or more TT surgeries in the potential study eye/eyes.

Persons without TT were community members without trichiasis or history of trichiasis that were location-, sex-, and age-matched (within 5 years interval) to the corresponding persons with TT. These were selected in probability sampling technique. To select the community members, the research teams visited the sub-villages (30–50 households) of the identified persons with TT. A list of all potentially eligible people living in the sub-village of the person with TT was compiled with the help of the sub-village administrator and or the community health worker. One person was randomly selected from this list using a lottery method. The selectee underwent quick triaging to confirm that s/he didn't have (i) trichiasis, (ii) a history of trichiasis or trichiasis surgery, and (iii) history of trichiasis in their family.

The experts involved in the conceptual, semantic, and item equivalence measurement were selected purposively. A balance based on their field of expertise in mental health, social science, linguistics, or trachoma; their experience in similar research; and their likely availability to contribute were sought during the selection process. The research data collectors involved in the operational equivalence were health professionals selected based on their previous experience in similar research.

## Scales

The 5-QSI (Table 1) has two 5-item scales developed to assess and monitor NTD-related stigma in both the community and affected persons. The 5-Question Stigma Indicator–community stigma (5-QSI-CS) was developed to assess and monitor community stigma towards a person affected with an NTD; and the 5-Question Stigma Indicator–affected person (5-QSI-AP) was developed to assess and monitor felt stigma in affected persons. Response scoring of each item in both 5-QSI-CS and 5-QSI-AP scales is 0 (never or I do not know), 1 (sometimes), or 2 (often/usually), giving a total score ranging from 0 to 10. The English version of the 5-QSI-CS has been validated in Ultra Pradesh, India, as part of the NTD toolkit to assess and monitor leprosy-related stigma [11]. The 5-QSI-AP has been validated in Nepal (i) for cross-cultural equivalence in Nepali in leprosy-affected people [10] and (ii) for internal consistency as part of a study conducted to determine factors influencing mental well-being among people affected with leprosy [12].

**Table 1.  5-Question Stigma Indicators (5-QSI) for monitoring and discrimination against persons affected by [condition].**

**(a) 5-Question Stigma Indicator–Community Stigma (5-QSI-CS)**

| Item # | In your community or neighbourhood | Never | Some-times | Often/Usually | Don't Know |
|---|---|---|---|---|---|
| 1 | Would having (had) [condition] cause problem for a person to find work or keep his or her job? | 0 | 1 | 2 | 0 |
| 2 | Would someone with [condition] be worried about others knowing this? | 0 | 1 | 2 | 0 |
| 3 | Does having [condition] cause shame to the person affected? | 0 | 1 | 2 | 0 |
| 4 | Would [condition] cause a problem for a person to get married or in an existing marriage? | 0 | 1 | 2 | 0 |
| 5 | Would people try to avoid someone with [condition]? | 0 | 1 | 2 | 0 |

**(b) 5-Question Stigma Indicator–Affected Persons (5-QSI-AP)**

| Item # | In the past year | Never | Some-times | Often/Usually | Don't Know |
|---|---|---|---|---|---|
| 1 | Have you experienced problems in finding or keeping work because you have (had) [condition]? | 0 | 1 | 2 | 0 |
| 2 | Have you been worried about others finding out you have (had) [condition]? | 0 | 1 | 2 | 0 |
| 3 | Have you felt ashamed because of your condition? | 0 | 1 | 2 | 0 |
| 4 | Have you had problems getting married / in your marriage because of [condition]? | 0 | 1 | 2 | 0 |
| 5 | Have people tried to avoid because you have (had) [condition]? | 0 | 1 | 2 | 0 |

5-QSI-CS indicator score: 0–10

5-QSI-AP indicator score: 0–10

The Social Distance Scale (SDS) has been employed to measure the construct validity of the 5-QSI-CS. It is one of the most used stigma tools that measures the social distance the interviewee wants to keep towards a person with a particular condition [16]. It can be used to assess the perceptions of the community towards people affected by NTDs by asking how they feel regarding different types of social relationships (e.g. neighbors, caregivers, colleagues). The SDS interview begins with the reading of a gender-specific vignette, a short description of a person with some typical characteristics of the condition (in the case of this study TT), followed by seven questions concerning the person in the vignette, with four response options: definitely willing (0 points), probably willing (1), probably not willing (2), and definitely not willing (3). The content of the two (male/female) vignettes is similar except for the gender, describing a male named Abebe or a female named Abebech, depending on the gender of the interviewee. The English and Amharic vignette versions were developed by EH and reviewed by ZT and FA. The English and Amharic versions of the SDS along with the vignette used in this study are attached in S1 and S2 Tables respectively.

## Sample size

For the qualitative exploratory component, we planned to enroll about 50 conveniently selected persons with TT and 50 persons without TT, with the final number to be determined by data saturation. In addition, we anticipated engaging about 20 experts and eight research data collectors. The sample size for the measurement equivalence was calculated based on the minimum number of participants required for a factor analysis. The rule of thumb is to use a minimum of two and a maximum of 20 people per item to estimate the sample size, but with a minimum number of 100 subjects to ensure stability of the variance-covariance matrix in factorial analysis [17, 18]. Based on this, using a 20:1 participant-to-item ratio, the required sample size was 100 persons with TT for the 5-QSI-AP and 100 TT-free community members for the 5-QSI-CS. Accounting for about 20% non-response, 120 persons with TT and 120 persons without TT were planned to be enrolled. For the repeatability component, the rule of thumb is at least 50 repeated measures are required. Accounting for 20% non-response we planned to re-assess 60 people from each group of persons with TT and without TT, randomly selected from the initial interviewee list.

## Cultural equivalence testing

The cross-cultural adaptation framework for this study has been illustrated in Fig 1.

## Qualitative exploratory component

The qualitative exploratory study was conducted to measure the conceptual, semantic, item, and operational validity of the 5-QSI scales as follows.

## Conceptual validity

Measures the way in which different populations conceptualize stigma and the value they place on the questions included to measure it [14]. Conceptual validity in this study was measured by assessing the local community's conceptualization of stigma, the appropriateness of the questions in the target population, and acceptance of the stigma scales by persons with TT and the community. A panel meeting was held to review the appropriateness of the scales and each of the items, with mental health, trachoma, social science, and linguistics experts. Items were rated as 'not relevant', 'somewhat relevant', or 'very relevant'. Items rated as 'not relevant' were proposed to be dropped, while those rated as 'somewhat relevant' were discussed in the panel.

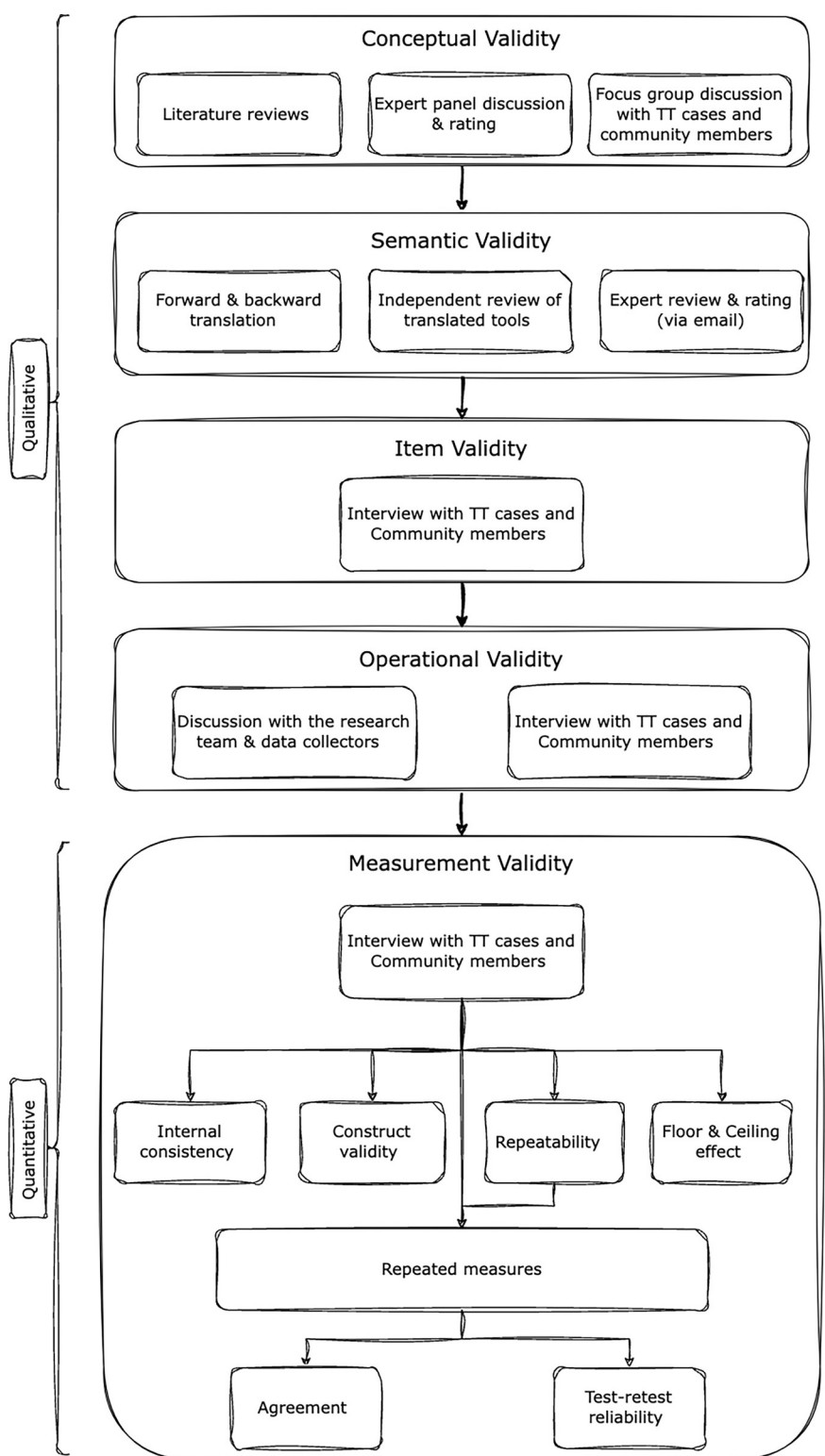

**Fig 1. Stigma tool cross-cultural adaptation framework.**

Focus Group Discussions (FGDs) were conducted with persons with and without TT to obtain deep insight about stigma experienced by affected people and community stigma. For the 5-QSI-AP two FGDs were conducted with persons with TT, and for the 5-QSI-CS two FGDs were conducted with persons without TT. A guide comprised of engagement, exploratory, and exit questions was used with an expert (FA) facilitating the focus group discussion and an assistant (EH) taking notes.

## Semantic validity

Refers to the transfer of meaning across languages without compromising or losing the original meaning [14]. Important aspects of semantic validity involve ensuring the level of language used is appropriate to the needs and culture of the target community [14]. To ensure semantic validity, we employed the following process. First, two mental health experts (professors at Addis Ababa University and the University of Gondar) with experience and expertise in translating similar tools were asked to translate the 5-QSI from the source language (English) to the target language (Amharic) using a predefined guide to standardize their efforts. They were specifically asked to consider the question "can the translated version be understood by a 12-year-old native speaker of the target language (Amharic)." Second, the two translated versions were reviewed by a third translator and consolidated to produce a single Amharic version. Third, the Amharic version was back-translated to English independently by two linguistics professors from Bahir Dar University who were not involved in stigma research. Fourth, two independent stigma experts who were not involved in the original translation process were asked to comment on the semantic equivalence of the back-translated versions. One of these experts was among the developers of the original 5-QSI tool. Fifth, the reviewed versions were sent back to the back-translators for revisions, after which a single back-translated version was produced. Sixth, the expert panel was emailed the translated versions to (a) review and rate the cultural equivalence of the translated tool (sufficiency) in a wide range of perspectives and identify and address residual discrepancies between the target and source tool, and (b) provide recommendations and approve a pre-final version of the Amharic 5-QSI scales. The forward translation and back-translation sufficiency rating was conducted in comparison with the source tool based on a predefined semantic and conceptual parameter, S3 Table. Each item forward and back translation was rated as 'adequate' (no revision needed), 'questionable' (needs discussion, some adjustment may be needed), or 'inadequate' (needs detailed special review, major revision may be needed). Finally, the pre-final version of the Amharic tool was validated by interviewing persons with TT (for the 5-QSI-AP) and persons without TT (for the 5-QSI-CS). The participants were asked to paraphrase the translated items to get insight into the similarity of meaning between the target (Amharic) and source (English) versions, and they were asked the question "What do you think this [question] intends to explore?" The participant responses were rated if the s/he clearly understood the meaning of the question or what it intended to explore.

## Item validity

Refers to how items are conceptualized and sampled and whether items are equally relevant and acceptable in the source and target cultures [14]. To measure item validity, the qualitative study participants were asked about the relevance and acceptability of items in the scales after its administration and were asked the questions: "Do you think this [question] is appropriate/acceptable to assess stigma in your community?", "If you think that this [question] is not acceptable to assess stigma in persons affected with TT, please tell us why and how it can be improved?", and "How relevant is this [question] to you to assess stigma in persons with TT?"

### Operational validity

Refers to the suitability of using similar questionnaire formats, instructions, mode of administration, and measurement methods in the target populations as was used in the original setting [14]. To measure operational validity, people with TT, community members, and data collectors were asked for clarity of questionnaire instruction, appropriateness of questionnaire format, mode of administration, and measurement method.

### Quantitative component

**Measurement equivalence.** Refers to the extent to which the psychometric properties, primarily in terms of their reliability, responsiveness, and construct validity, of different language versions of the same instrument are similar [14]. Measurement equivalence of the 5-QSI in this study was determined using data from 120 persons with TT and 120 persons without TT in terms of the following psychometric properties:

### Internal consistency

Refers to the extent to which the items in the questionnaire are homogeneous, thus measuring the same concept [14].This was evaluated using exploratory factor analysis and Cronbach's alpha coefficient ($\alpha$).

*Construct validity*. Refers to the extent to which the stigma tools behave as they are expected to behave [14].The construct validity of the 5-QSI-AP was measured in discriminant validity using known group difference criteria. The predefined hypothesis was that persons with more TT (higher number of lashes, trichiasis for longer period) and its sequelae (worst corneal opacity, reduced visual acuity) will have higher stigma scores than their counterparts. The construct validity for the 5-QSI-CS was measured using convergent validity tested through Spearman correlation between the scores of 5-QSI-CS and the SDS. The predefined hypothesis was that a moderate positive correlation (Spearman correlation coefficient between 0.4 and 0.8) exists between the 5 QSI-CS and the SDS stigma tools.

*Repeatability*. Assesses the degree to which repeated measurements provide similar answers [18]. About 60 persons with TT and 60 persons without TT enrolled in the measurement equivalence study were revisited about two weeks after the initial interview to be asked the same questions by the same interviewers.

*Floor and ceiling effects*. Floor and ceiling effects measure the tendency of respondents to choose the extremes of a scale in their responses. In this study, floor and ceiling effects were considered present in the 5 QSI-AP and the 5 QSI-CS if >15% of the respondents had the lowest or highest possible score respectively.

### Data collection

Data were collected by eight experienced data collectors, nurses, and health officers. Data collectors received detailed theoretical (total 10 days) and practical training (7 days) on the WHO mental health Gap Action Program (mhGAP), the objective of the study, the data collection tools, standard operating procedures (SOP), research ethics, and good clinical practice (GCP). The practical training included ensuring that data collectors administer the questions in the field uniformly following standardized instructions. The data collectors were standardized for TT and corneal opacity diagnosis against the gradings of EH and had a strong agreement ($\kappa$>0.70).

All questions were interviewer administered. In addition to collecting data using the scales, data on socio-demographic variables (age, gender, educational level, marital and socio-

economic status) were collected. A brief ophthalmic questionnaire was used to elicit information on the duration of trichiasis, pain from trichiasis, and other trichiasis-related concerns from persons with TT. Detailed instructions on how to conduct the interview were described in the SOP document, which then was given to each data collector. The same standardized instructions were read to all participants and informed on what the questions were about, the time frame they constituted, and how to answer, including giving the best answer they could when they were not sure about how to answer. If a participant failed to understand a question, the question was repeated once. All interviews were conducted in private. The interviewer was not permitted to discuss the questions or answers of the participant during the interview session. Once the interview was completed, persons with TT underwent an ocular examination to grade the severity of their trichiasis and the presence of other blinding conditions. The SDS along with its vignette was administered to community members after the 5-QSI-CS had been administered to avoid response bias on the 5-QSI-CS due to the vignette.

FGDs were audio recorded and stored in a password-protected computer. The item and semantic validation data were collected on paper forms, while the measurement validity and clinical data were collected using an electronic Open Data Kit (ODK) form from which data were uploaded daily to a secured server at the London School of Hygiene & Tropical Medicine, UK. Data from the paper forms were double-entered in a Microsoft Access database in a password-protected computer in the project office.

## Data analysis

Data were checked for completeness, missing values, and questionnaire coding. Qualitative data were thematically analyzed using Open Code 4.03 after being transcribed and translated into English. Key concepts were directly quoted in the results. Quantitative data collected to assess measurement validity were exported to STATA/SE 17.0 and analyzed using the following methods.

The internal consistency of items was measured using Exploratory Factor Analysis (EFA) and Cronbach's alpha coefficient. EFA was employed to measure the underlying relationship between the items and constructs (factors) measured by the scale. Prior to EFA, factorability of the correlation matrix was determined using Bartlett's test of sphericity and the Kaiser–Meyer–Olkin (KMO) measure of sampling adequacy. The Bartlett's test of sphericity was used to check if the correlation matrix was not random (null hypothesis that the variables are orthogonal or uncorrelated) at a 0.05 significance level. A KMO of $>0.50$ was used to determine sample adequacy. Cronbach's alpha coefficient values between $\geq 0.70$ and $\leq 0.95$ were classified as acceptable.

Construct validity was measured for the 5-QSI-AP using stigma score differences between persons with more severe TT (higher number of lashes and trichiasis for a longer period) and its sequelae (corneal opacity and vision impairment) and their counterparts (discriminant validity) in a Wilcoxon Rank-Sum test, as the data was not normally distributed. TT severity was categorized as minor ($\leq 5$ eyelashes touching the eye or evidence of epilation in $\leq 1/3$ of the eyelid) and major ($>5$ eyelashes touching the eye or evidence of epilation in $>1/3$ of the eyelid) in the worst affected eye. The 5-QSI-CS construct validity was measured by examining the association between the scores of the 5-QSI-CS and SDS using the Spearman correlation coefficient.

Test-retest reliability was measured by comparing the initial and the repeat data to determine linear relationship using Intraclass Correlation Coefficients and their 95% confidence intervals (CIs). A coefficient of at least 0.50 was used as a cutoff for evidence of moderate correlation or reliability. Presence of any systematic differences in mean stigma scores between

the initial and the repeat measure were assessed using Wilcoxon signed-rank test. The Bland-Altman method was employed from the individual mean stigma scores and their differences between the initial and repeat assessments to determine how close the scores on repeated measures were. The 95% CI limits of agreement were calculated as the mean stigma score difference between the two measures ±1.96 multiplied by the standard deviation (SD) of the mean differences.

## Ethical considerations

Ethical approval was sought from the Ethics Review Committees of the Amhara Public Health Institute in Ethiopia (approval number H/ R/T/Tin/3/756), the London School of Hygiene & Tropical Medicine in the UK (approval number 16277), and Bahir Dar University College of Health and Medical Science (IRB decision number 003). The study was conducted in compliance with the Declaration of Helsinki and ICH-GCP. Verbal consent was taken from experts involved in the conceptual equivalence process. Written informed consent was obtained from persons with and without TT. If the participant was unable to read and write, the information sheet and consent form were read to them and their consent recorded by thumbprint, in the presence of a witness. Interviews were conducted privately. Persons with TT were offered free surgical management immediately after the data collection was completed. Those who declined surgery were offered epilation and provided the details of where to get surgical management anytime they needed to.

## Results

### Study participants

The data collection for this study was completed between the 1st of August and 18th of December 2020. A total of 390 people participated (138 male and 252 female) including 181 persons with TT; 182 persons without TT; 19 mental health, trachoma, social science, and linguistics experts who participated in the translation, review, and panel discussion of the stigma tools; and 8 research data collectors (for operational validity). The study activities and participants for each are summarized in Table 2.

### Qualitative study

**Conceptual equivalence.** The conceptual equivalence was mainly evaluated through an expert panel discussion and FGDs. Four FGDs were conducted with women with TT, women without TT, men with TT and men without TT, each involving six people. The mean age of the FGD participants was 44.1 (SD: 18.1). Most of them were farmers (19/25, 76.0%) and didn't attend formal education (23/25, 92.0%).

The first step in the conceptual equivalence exercise was to determine the most appropriate Amharic term that effectively defines "trachomatous trichiasis." Several Amharic phrases were suggested during the panel discussion which directly translate to "trachomatous eye illness", "trachomatous eye stubbing from eyelashes," and "eye hair growing," for example. It was noted that sometimes trachoma is a blanket term for all kinds of itchy conjunctival eye diseases, therefore, it's important to be specific to effectively describe the blinding stages of trachoma. It was agreed that the appropriate term in the community needed to be explored along with a brief description of the condition supported with photographs.

During the FGDs with community members, the participants were shown pictures of eyes with varying severities of TT and were asked for the term commonly used in their community to describe this condition. The most commonly used term to describe TT in the study

**Table 2. Summary of study activities and participants.**

| Cultural Equivalence Test Components | Data collection Activities | Participants | | | Output |
|---|---|---|---|---|---|
| | | Male n (%) | Female n (%) | Total N | |
| Conceptual validity | Literature reviews | - | - | - | Concepts of stigma elicited and documented |
| | Expert panel discussion and rating | 12 (100) | 0 | 12 | Appropriateness/relevance of the tools and each of their items discussed and rated (content validity rating) |
| | Focus Group Discussion with TT cases (2 sessions) | 6 (50.0%) | 6 (50.0%) | 12 | Data on the beliefs of the community on stigma gathered |
| | Focus Group Discussion with community (2 sessions) | 6 (46.1%) | 7 (53.8%) | 13 | |
| Semantic Validity | Forward and backward translation of tools by experts | 4 (100%) | 0 | 4 | Ensured the transfer of meaning of each item without compromising or losing original meaning |
| | Independent reviews of translated tools by experts | 1 (33.3%) | 2 (66.7%) | 3 | Reviewed and commented on the semantic equivalence of tools during the forward and backward translation |
| | Expert review and rating (via email) | 3 (100%) | 0 | 3 | The "sufficiency" of tools translation rated |
| Item validity | Interviewed TT cases | 10 (31.2%) | 22 (68.7%) | 32 | Data on the understandability, relevance and acceptance of the stigma and suicidality tool items collected |
| | Interviewed Community members | 9 (28.1%) | 23 (71.9%) | 32 | Data on the understandability, relevance and acceptance of the stigma and suicidality tool items collected |
| Operational Validity | Discussion with research team | 8 (100%) | 0 | 8 | Instruction and mode of administration tested, and tools finalized |
| | Interviewed TT cases | 5 (29.4%) | 12 (70.6%) | 17 | |
| | Interviewed community members | 7 (41.2%) | 10 (58.8%) | 17 | |
| Measurement validity | Interviewed TT cases | 35 (29.2%) | 85 (70.8%) | 120 | Data on the psychometric properties of the tools collected |
| | Interviewed community members | 35 (29.2%) | 85 (70.8%) | 120 | |
| Reproducibility | Repeated measures in TT cases | 16 (26.7%) | 44 (73.3%) | 60 | Data on test-retest agreement collected |
| | Repeated measures in community | 16 (26.7%) | 44 (73.3%) | 60 | |
| **Total (excluding repeated measures)** | | 138 (35.4%) | 252 (64.6%) | 390 (100%) | |

community was a three-word Amharic phrase that directly translates into "eye hair sprouting" or "eye hair growing." As predicted in the expert panel discussion, there was a tendency to relate the word "trachoma" to any inflammatory eye disease characterized by inflammation, redness, and discharge but not to "trichiasis."

The next step was to determine the relevance of the tool and the items in the target culture. We have found evidence that the tool and the items are relevant and important in this Amharic-speaking study community. There were no indications that led us to change the emphasis placed on any item.

The expert panel agreed that most of the items included in both 5-QSI-AP and 5-QSI-CS are relevant except for item two, "worried about others finding out" they have the condition, which was rated as "somewhat relevant." The issue raised on item two by the panel of experts was that most people with TT in the study community would rather ask other people to epilate their trichiatic eyelashes than hide. However, the community FGDs revealed that may not always be the case, suggesting that often persons with TT in the community don't want others to know that they have trichiasis.

The following quotes support the relevance of item two and item three on both perceived worry and shame:

(. . .) *"There are some individuals who say that they will be sick when they see a reddish and swollen eye. (. . .) when we meet such people who fear they will be sick by looking on a sick reddish eye, we don't confront them. Usually, people. . .just look down on the ground for fear of other finding their condition."* (FGD1, Merawi, 49-year- old male with epilated TT)

(. . .) *"No one loves you if you lose your eyesight. No one takes care of me and comes around me if I become blind and unable to work."* (FGD3, Merawi, 30-year-old female with TT)

(. . .) *"Sometimes teasing follows (. . .) sometimes people insult our kids by saying the name of our disease (. . .) the insult is you full of jelly like [discharge]". . .* (FGD3, Merawi, 38-year-old female with TT)

(. . .) *"There is much marginalization and worse still there are some people who insults them by saying blind (. . .) there are some people who even insult the kids as a son of a blind even if the kids are healthy enough."* (FGD2, Merawi, 81-year-old male without TT)

The next quotes from the community members elaborate on the problems people with TT face in getting married or in ongoing marriage, which is explored in item four of the 5-QSI:

(. . .) *"There are many who got troubles from their spouses. Their husbands frequently insult them and express their fears that one day they will catch the disease themselves and finally they divorce."* (FGD 4, Merawi, 20-year-old female without TT)

(. . .) *"As to me, how can I marry a man with such difficult condition [TT]? What kind of life can we possibly have in such condition? Therefore, I will demand a divorce."* (FGD 4, Merawi, 30-year-old female without TT)

(. . .) *"Unless she is his first wife or she has already got some other kids, no one is willing to marry such a man."* (FGD2, Merawi, 40-year-old male without TT)

Item five explores whether people affected by the condition experience avoidance. The FGDs indicated that people with TT are often avoided by other people in their community. The following quotes elaborate on the avoidances community members show toward people with TT:

(. . .) *"The eyes will have lots of white jelly like cream and many tears. (. . .) people do not feel good about such people. People do not consider them as fellow human beings. People do not eat with him and they do not want them to come any closer. (. . ..) There are also some God fearing and good people who are willing to be served from the same meal. There are also very arrogant people who would focus their eyes on the patient rather than the meal."* (FGD2, Merawi, 75-year-old priest without TT)

(. . .) *"If I see him rub his eyes and invite me to eat together with him, I refuse to do so (. . .) even if he insists, I will never eat together with this man."* (FGD2, 81-year-old, male without TT)

(. . .) *"It is not known how the disease comes, so it is better to be far from person with this disease (. . .) I don't take in food from him [a person with TT]."* (FGD4, Merawi, 30-year-old female without TT)

*(. . .) "There is one old woman [with TT]. When I go to her home and she give me some injera, I refuse to eat it, and if she happens to have made some coffee, I won't drink it either. I tell her I will never drink from a cup you have prepared. She becomes very sad and deplores God for giving her this disease. I always help her by taking chores for her, but I will never eat any food given to me by her." (FGD4, Merawi, 24-year-old female without TT)*

**Semantic, item, and operational equivalence.**   *Participants*. A total of 32 persons with TT were involved in in-depth interviews (IDIs) to assess semantic and item equivalences of 5-QSI-AP. The majority were female (22, 68.7%). Their mean age was 48.7 (SD: 12.7). Most of them were farmers (15, 46.8%) and didn't attend formal education (20, 62.5%). A separate group of 32 persons without TT was also involved in IDIs to assess semantic and item equivalences of the 5-QSI-CP. The majority were again female (23, 71.9%). Their mean age was 35.9 (SD: 13.4). Half of them were farmers (16, 50.0%) and didn't attend formal education (17, 53.1%). Operational equivalence was assessed with eight data collectors (interviewers), 17 persons with TT, and 17 persons without TT.

*Semantic equivalence*. Following the forward and backward translation sufficiency rating, all items were rated "adequate" except for item three of the 5-QSI-CS (Does having [condition] cause shame to the person affected) and item five for both the 5-QSI-CS (Would people try to avoid someone with [condition]?) and 5-QSI-AP (Have people tried to avoid because you have (had) [condition]?). In Item three the word "victim" was used in the back translation due to the use of a term that also translates as victim in the forward translation, which was later revised. For item number five the term "avoid" was forward translated into an Amharic term that back translated into "discriminate." This was because the term "avoid" was less contextually understood in the community than "discriminate," a term that pertains to a broader concept and which has been used for several years in the community in relation to the HIV/AIDs prevention campaigns. Finally, item five was revised with an Amharic term that translates into "avoid" following a semantic validity exercise in the community.

Of the 32 persons with TT interviewed to test semantic validity, seven could not understand item one (Have you experienced problems in finding or keeping work because you have [condition]?) and another five understood it partially. Some participants linked this question with the concept of "*finding it hard to work because of their eye problem.*" Quotes below illustrate the ways respondents tended to relate to item one:

*(. . .) "The question says, can a person with [hair sprouting into the eye] work as a normal person does?" (IDI-1, 45-year-old female with TT)*

*(. . .) "The question says, 'do you find it hard to work because of your eye problem?" "This question talks about the hardship on work because of my eye problem." (IDI-1, 35-year-old female with TT)*

*(. . .) "The question says, do you find problem to do work because [hair sprouted] into the eye?" "[Hair sprouted/grown into the eye] cause problem to stay in a work." (IDI-1, 55-year-old female with TT)*

Similarly, in interviews with persons without TT, all questions were understood except item one of the 5-QSI-CS by three out of 32 people. Quotes below illustrate the ways these respondents tended to relate to item one:

*(. . .) "The question says don't people find difficulty to do work?" (IDI-2, 25-year-old female without TT)*

(...) *"The question says do people find difficulty to do their daily work?" "Do a person with [hair grown in eye] find difficulty to stay at or find work, as it reduces their vision?" (IDI-2, 30-year-old female without TT)*

(...) *"They can't work as a healthy person as their vision is weakened to do so." (IDI-2, 25-year-old female without TT)*

*Item equivalence.* All items in 5-QSI-CS scale were stressed by persons without TT as relevant and appropriate to explore the intended phenomenon (community stigma). A similar rating was given for 5-QSI-AP by persons with TT except for item one which was rated as "less relevant" by the majority of the respondents. The issue raised on item one was that most of the TT-affected people were farmers who work on their own farms and that measuring the impact of the condition as "keeping" or "maintaining" work could be less relevant. To address this issue the Amharic translation of item one was revised into something that back translates into English as "Have people tried to not offer you work or not to work with you because you have (had) [trachomatous trichiasis]?" for the 5-QSI-AP and "Would people try not to offer work or prevent someone with [trachomatous trichiasis] from working with them?" for the 5-QSI-CS, which were clearly understood and deemed relevant by persons with and without TT respectively in subsequent interviews. The revised English version of the 5-QSI is provided in Table 3. The revised Amharic version of the 5-QSI is provided in Fig 2.

*Operational equivalence.* After testing the prefinal version of the scale on persons with and without TT, the interviewers recommended a modification to be made on the format or sequences of items. The interviewers recommended items should be administered in order of difficulty being understood by the study community, from easy to difficult. Therefore, based on the input from interviewers the items were reordered as follows: item three to be asked first, followed by item two, item four, item one, and finally item five. The measurement equivalence of the scales was assessed on the revised versions, Table 3. No other changes were recommended.

**Table 3. 5-QSI with revisions on item one and administration ordering (English).**

**5-QSI-CS**

| Revised item order | Original item order | In your community or neighborhood |
|---|---|---|
| 1 | 3 | Does having [trachomatous trichiasis] cause shame to the person affected? |
| 2 | 2 | Would someone with [trachomatous trichiasis] be worried about others knowing this? |
| 3 | 4 | Would [trachomatous trichiasis] cause a problem for a person to get married or in an existing marriage? |
| 4 | 1 | Would people try not to offer work or prevent someone with [trachomatous trichiasis] from working with them? |
| 5 | 5 | Would people try to avoid someone with [trachomatous trichiasis]? |

**5-QSI-AP**

| Revised item order | Original item order | In the past year |
|---|---|---|
| 1 | 3 | Have you felt ashamed because of the [trachomatous trichiasis]? |
| 2 | 2 | Have you been worried about others finding out you have (had) [trachomatous trichiasis]? |
| 3 | 4 | Have you had problems getting married / in your marriage because of [trachomatous trichiasis]? |
| 4 | 1 | Have people tried to not to offer work or prevent you from working with them because you have (had) [trachomatous trichiasis]? |
| 5 | 5 | Have people tried to avoid because you have (had) [trachomatous trichiasis]? |

| 5-QSI-CS | | |
|---|---|---|
| **Revised item order** | **Original item order** | እርስዎ በሚኖሩበት ማህበረሰብ ወይም አካባቢ |
| 1 | 3 | [የአይን ፀር መብቀል] በሰውየው ላይ የሀፍረት ስሜት ይፈጥራል? |
| 2 | 2 | [የአይን ፀር የበቀለበት] ሰው ሌሎች ሰዎች ስለህመሙ አንዳያወቁበት ሊጨነቅ ይችላል? |
| 3 | 4 | [የአይን ፀር መብቀል] ግለሰቡ ትዳር አንዳይመሰርት ወይም ያገባ ከሆነ በትዳሩ ላይ ችግር ሊፈጥር ይችላል? |
| 4 | 1 | ሰዎች [የአይን ፀር የበቀለባትን] ሰው ስራ ሊነፍጉት ወይንም አብረውቸው አንዳይሰሩ ሊያደርጉት የችላሉ? |
| 5 | 5 | ሰዎች [የአይን ፀር የበቀለበትን] ሰው ሊያርቁት ይችላሉ? |
| **5-QSI-AP** | | |
| **የተሻሻለ ቅደም ተከተል** | **የመጀመሪያው ቅደም ተከተል** | ባለፈው አንድ አመት |
| 1 | 3 | [የአይን ፀር መብቀል] ስላለብዎት የሀፍረት ስሜት ተሰምቶዎት ያውቃል? |
| 2 | 2 | [የአይን ፀር መብቀል] አንዳለብዎት (አንደነበረብዎት) ሌሎች ሰዎች አንዳያውቁ ተተጫጭቀው ያውቃሉ? |
| 3 | 4 | [የአይን ፀር መብቀል] ስላለብዎት/ስለነበረብዎት ትዳር ለመመስረት ተቸግረዋል/ ያገቡ ከሆነ በትዳርዎ ላይ ችግር ገጥሟዎት ያውቃል? |
| 4 | 1 | [የአይን ፀር መብቀል] ስላለብዎት ሰዎች ዕርስዎን ሥራ ላለማሰራት ወይንም ከዕረስዎ ጋር አብሮ ስራ ላለመስራት ሞክረው ያውቃሉ? |
| 5 | 5 | [የአይን ፀር መብቀል] ስላለብዎት (ስለነበረብዎት) ሰዎች ሊርቁዎት ሞክረው ያውቃሉ? |

**Fig 2. 5-QSI with revisions on item one and administration ordering (Amharic).**

## Quantitative study

**Measurement equivalences.** *Sociodemographic characteristic.* The sociodemographic and clinical characteristics of the measurement equivalence assessment participants are presented in Table 4. A total of 120 persons with TT were enrolled for the 5-QSI-AP. The majority were female (70.8%). Their mean age was 46.5 (SD: 13.0). Most of them were farmers (68.3%), married (70.0%), didn't attend formal education (84.2%), and had major TT (63.3%). For the 5-QSI-CS, 120 community members were enrolled. Their mean age was 45.7 (SD: 13.4). The majority were female (70.8%), married (75.8%), and didn't attend formal education (72.3%).

*Item characteristics.* The stigma scores for the 5-QSI-CS and the 5-QSI-AP ranged from 0–9 and 0–7 respectively, Fig 3. Descriptive statistics for each item are presented in Table 5. The mean total scores for the 5-QSI-CS and 5-QSI-AP were 2.8 (SD: 2.2) and 2.10 (SD: 1.8) respectively. Items five (measures avoidance) and one (measures problems working with other people) had the lowest average score.

*Internal consistency.* KMO analysis showed that both the 5-QSI-CS (KMO = 0.63) and 5-QSI-AP (KMO = 0.54) samples are adequate for factorial analysis. The Bartlett's test showed that the items in both the 5-QSI-CS scale and the 5-QSI-AP scale were fit as one-dimensional, (p<0.001). Exploratory factor analysis showed that factor 1 explains 88% and 87% of the variance in the data for 5-QSI-CS and 5-QSI-AP respectively.

The 5-QSI-CS showed acceptable internal consistency with Cronbach's α of 0.70, while the internal consistency of 5-QSI-AP was limited with a Cronbach's α of 0.57. The SDS showed Cronbach's α of 0.88 with consistent item-scale correlation ranging between 0.66 and 0.86.

The reliability statistics for the 5-QSI-CS and the 5-QSI-AP are presented in Table 6.

The factor loadings for each item in the 5-QSI-CS and 5-QSI-AP are presented in Table 7. For the 5-QSI-CS items one, three, four, and five load on factor 1, while only item two (worry about others knowing) loads on factor 2. This suggests that item two may measure a slightly different concept than the other items in the scale. This is particularly justified with the reliability statistics results in Table 6 in which Item two in the 5-QSI-CS, has the lowest item-scale (0.54) and item-rest (0.25) correlation. Item-scale correlation indicates the correlation between an item and the scale that is formed by all items with that item included in the test, while item-rest correlation indicates the correlation between an item and the scale that is formed by all other items without that item. Removing item two from the scale increased the reliability coefficient (Cronbach's α) of the 5-QSI-CS to 0.73, suggesting that the 5-QSI-CS is better off as a four-item scale.

**Table 4. Sociodemographic and clinical characteristics measurement equivalence participants.**

| Variables | TT Cases (5-QSI-AP) | | Community Members (5-QSI-CS) | |
|---|---|---|---|---|
| | n / 120 | (%) | n / 120 | (%) |
| **Age in years, mean (SD)** | 46.5 | (13.0) | 45.7 | (13.4) |
| **Gender, female** | 85 | (70.8) | 85 | (70.8) |
| **Illiterate** | 101 | (84.2) | 87 | (72.5) |
| **Marital status** | | | | |
| Married | 84 | (70.0) | 91 | (75.8) |
| Widowed | 19 | (15.8) | 15 | (12.5) |
| Divorced | 15 | (12.5) | 14 | (11.7) |
| Single | 2 | (1.7) | 0 | (0.0) |
| **Level of education** | | | | |
| No formal education | 101 | (84.2) | 88 | (73.3) |
| Religious or other forms of education (can read and write) | 5 | (4.2) | 8 | (6.7) |
| Some primary school (Grade 1–8) | 10 | (8.3) | 17 | (14.2) |
| High school education | 0 | (0.0) | 1 | (0.8) |
| College/ technic school diploma | 3 | (2.5) | 4 | (3.3) |
| Degree and above | 1 | 1 (0.8) | 2 | (1.7) |
| **Job** | | | | |
| Farmer | 82 | (68.3) | 76 | (63.3) |
| Housewife | 10 | (8.3) | 13 | (10.8) |
| Employed/self employed | 8 | (6.7) | 11 | (9.2) |
| Daily labourer | 16 | (13.3) | 17 | (14.2) |
| No job (Students, retired) | 4 | (3.3) | 3 | (2.5) |
| **Position in the community** | | | | |
| Ordinary member (no unique role) | 112 | (93.3) | 107 | (89.2) |
| Village administration team member | 5 | (4.2) | 11 | (9.2) |
| Religious leader (Pries or similar) | 2 | (1.7) | 1 | (0.8) |
| Community health worker | 0 | (0.0) | 1 | (0.8) |
| Teacher | 1 | (0.8) | 0 | (0.0) |
| **Self-rated wealth** | | | | |
| Very wealthy/ Wealthy | 9 | (7.5) | 9 | (7.5) |
| Middle | 62 | (51.7) | 87 | (72.8) |
| Very Poor / Poor | 49 | (40.8) | 24 | (20.0) |
| **TT duration** | | | | |
| ≤5years | 60 | (50.0) | | |
| >5 years | 60 | (50.0) | | |
| **TT severity in the worst affected eye** | | | | |
| Minor TT[a] | 44 | (36.7) | | |
| Major TT[b] | 76 | (63.3) | | |
| **Corneal opacity in the worst affected eye** | | | | |
| No or peripheral corneal opacity (C0/C1) | 51 | (42.5) | | |
| Central corneal opacity (C2a –Phthisis) | 69 | (27.5) | | |
| **Visual acuity in the better eye** | | | | |
| No vision impairment (≥6/18) | 97 | (80.8) | | |
| Vision impairment (<6/18) | 23 | (19.2) | | |

[a] defined as ≤5 eyelashes touching the eye or evidence of epilation in <1/3$^{rd}$ of the eyelid.

[b] defined as .5 eyelashes touching the eye or evidence of epilation in ≥1/3$^{rd}$ of the eyelid.

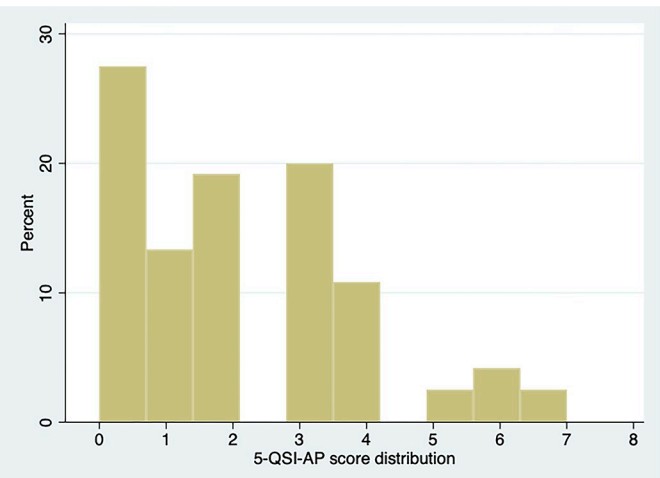
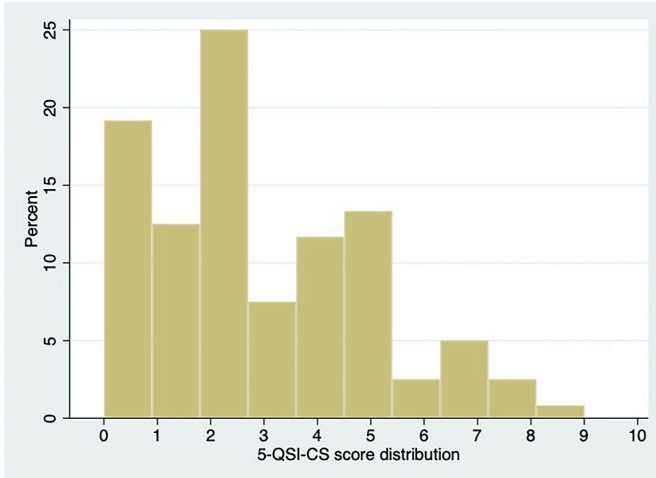

**Fig 3. Stigma score distribution for the 5-QSI-AP and the 5-QSI-CS.**

The factor loadings for the 5-QSI-AP supported two factors with items three, four & five loading on factor 1 while items one & two load on factor 2. These suggest that items one (problem working with other people) and two (worry about others knowing) may measure a slightly different concept than the other three items in the scale. However, the reliability statistics data in Table 6 indicates that removing any of the item and particularly items one and two from the scale resulted reduction in α to 0.50 and 0.51 respectively, suggesting a well-fitting items and that the 5-QSI-AP is suitable as a 5-item scale.

## Construct validity

Construct validity 5-QSI-AP was measured using discriminant validity. Supporting the predefined hypothesis, persons with more severe TT and its sequelae had higher mean stigma scores

**Table 5. Descriptive statistics of 5-QSI-CS and 5-QSI-AP.**

| 5-QSI-CS (score 0–2) (N = 120) | | | | |
|---|---|---|---|---|
| **Revised item order** | **Original item order** | **In your community or neighbourhood** | **Mean** | **SD** |
| 1 | 3 | Does having [trachomatous trichiasis] cause shame to the person affected? | 0.75 | 0.70 |
| 2 | 2 | Would someone with [trachomatous trichiasis] be worried about others knowing this? | 0.53 | 0.72 |
| 3 | 4 | Would [trachomatous trichiasis] cause a problem for a person to get married or in an existing marriage? | 0.73 | 0.70 |
| 4 | 1 | Would people try not to offer work or prevent someone with [trachomatous trichiasis] from working with them? | 0.37 | 0.58 |
| 5 | 5 | Would people try to avoid someone with [trachomatous trichiasis]? | 0.38 | 0.64 |
| | | Total Score | 2.8 | 2.2 |
| 5-QSI-AP (score 0–2) (N = 120) | | | | |
| **Revised item order** | **Original item order** | **In the past year** | **Mean** | **SD** |
| 1 | 3 | Have you felt ashamed because of the [trachomatous trichiasis]? | 0.91 | 0.78 |
| 2 | 2 | Have you been worried about others finding out you have (had) [trachomatous trichiasis]? | 0.64 | 0.81 |
| 3 | 4 | Have you had problems getting married / in your marriage because of [trachomatous trichiasis]? | 0.22 | 0.50 |
| 4 | 1 | Have people tried to not to offer work or prevent you from working with them because you have (had) [trachomatous trichiasis]? | 0.2 | 0.46 |
| 5 | 5 | Have people tried to avoid because you have (had) [trachomatous trichiasis]? | 0.13 | 0.36 |
| | | Total Score | 2.1 | 1.8 |

**Table 6. Reliability statistics for 5-QSI-CP and 5-QSI-AP.**

**5-QSI-CS reliability statistics**

| Revised item order | Original item order | In your community or neighborhood | Item-scale correlation[c] | Item-rest correlation[d] | Cronbach's Alpha if Item Deleted |
|---|---|---|---|---|---|
| 1 | 3 | Does having [trachomatous trichiasis] cause shame to the person affected? | 0.75 | 0.55 | 0.60 |
| 2 | 2 | Would someone with [trachomatous trichiasis] be worried about others knowing this? | 0.54 | 0.25 | 0.73 |
| 3 | 4 | Would [trachomatous trichiasis] cause a problem for a person to get married or in an existing marriage? | 0.66 | 0.42 | 0.66 |
| 4 | 1 | Would people try not to offer work or prevent someone with [trachomatous trichiasis] from working with them? | 0.73 | 0.56 | 0.61 |
| 5 | 5 | Would people try to avoid someone with [trachomatous trichiasis]? | 0.71 | 0.52 | 0.62 |
| **Scale Cronbach's Alpha** | | | | | **0.70** |

**5-QSI-AP reliability statistics**

| Revised item order | Original item order | In the past year | Item-scale correlation[c] | Item-rest correlation[d] | Cronbach's Alpha if Item Deleted |
|---|---|---|---|---|---|
| 1 | 3 | Have you felt ashamed because of the [trachomatous trichiasis]? | 0.76 | 0.46 | 0.42 |
| 2 | 2 | Have you been worried about others finding out you have (had) [trachomatous trichiasis]? | 0.70 | 0.35 | 0.51 |
| 3 | 4 | Have you had problems getting married / in your marriage because of [trachomatous trichiasis]? | 0.54 | 0.30 | 0.53 |
| 4 | 1 | Have people tried to not to offer work or prevent you from working with them because you have (had) [trachomatous trichiasis]? | 0.57 | 0.36 | 0.50 |
| 5 | 5 | Have people tried to avoid because you have (had) [trachomatous trichiasis]? | 0.42 | 0.23 | 0.56 |
| **Scale Cronbach's Alpha** | | | | | **0.57** |

[c] Measures the correlation between the item and the scale formed by all items including that particular item. Likely to be distorted by poorly correlated or fitting item.
[d] Measures the correlation between the item and the scale formed by all other items excluding that particular item.

**Table 7. Exploratory factor analysis of 5-QSI-CS and 5-QSI-AP.**

**5-QSI-CS (N = 120)**

| Revised item order | Original item order | In your community or neighborhood | Factor 1 | Factor 2 |
|---|---|---|---|---|
| 1 | 3 | Does having [trachomatous trichiasis] cause shame to the person affected? | 0.60 | |
| 2 | 2 | Would someone with [trachomatous trichiasis] be worried about others knowing this? | 0.60 | |
| 3 | 4 | Would [trachomatous trichiasis] cause a problem for a person to get married or in an existing marriage? | | 0.69 |
| 4 | 1 | Would people try not to offer work or prevent someone with [trachomatous trichiasis] from working with them? | 0.59 | |
| 5 | 5 | Would people try to avoid someone with [trachomatous trichiasis]? | 0.69 | |
| Eigenvalue | | | 1.8 | 0.70 |

**5-QSI-AP (N = 120)**

| Revised item order | Original item order | In the past year | Factor 1 | Factor 2 |
|---|---|---|---|---|
| 1 | 3 | Have you felt ashamed because of the [trachomatous trichiasis]? | 0.39 | |
| 2 | 2 | Have you been worried about others finding out you have (had) [trachomatous trichiasis]? | | 0.48 |
| 3 | 4 | Have you had problems getting married / in your marriage because of [trachomatous trichiasis]? | 0.69 | |
| 4 | 1 | Have people tried to not to offer work or prevent you from working with them because you have (had) [trachomatous trichiasis]? | | 0.45 |
| 5 | 5 | Have people tried to avoid because you have (had) [trachomatous trichiasis]? | 0.55 | |
| Eigenvalue | | | 1.2 | 0.65 |

than their counterparts. Those with central corneal opacity had significantly higher stigma scores than those with no or peripheral corneal opacity (mean stigma score: 2.48 vs 1.59, p = 0.018). In addition, although it was not statistically significant, those with major TT in the most affected eye (mean stigma score: 2.26 vs 1.82, p = 0.22), ≥5 years TT duration (mean stigma score: 2.32 vs 1.88, p = 0.14), and vision impairment (mean stigma score: 2.61 vs 1.98, p = 0.29) had higher stigma scores than their counterparts. Construct validity for the 5-QSI-CS was measured using convergent validity assessing its correlation with the SDS using Spearman correlation. This has shown a weak correlation (r = 0.23).

## Repeatability

The 5-QSI-AP and 5-QSI-CS were re-administered after two weeks (median 13 days, range 9–14 days) on 60 persons with TT and 60 persons without TT respectively. The average age of persons with TT enrolled in the repeated measure was 45.4 years (SD: 13.4) and 73% were females. The average age of the persons without TT enrolled in the repeated measure was 44.3 years (SD: 13.7) and 73.3% were females.

The test-retest characteristics of the 5-QSI and the SDS as a comparator for the 5-QSI-CS are presented in Table 8. Both the 5-QSI-AP and the 5-QSI-CS had shown adequate test-retest reliability with a correlation of ≥0.60 and 0.53, respectively, and no evidence of systematic bias in mean stigma scores between the initial and the repeat measure for both the 5-QSI-AP (mean difference: 0.22, 95% CI: -0.23–0.67, p = 0.34) and the 5-QSI-CS (mean difference: -0.25, 95% CI: -0.88–0.38, p = 0.43). On the other hand, the SDS showed excellent test-retest reliability with a ≥0.70 correlation coefficient but with some evidence of bias between the initial and the repeat assessment (mean difference: -1.67, 95% CI: -2.98 - -0.36, p = 0.013). The 95% limits of agreement in the Bland-Altman analysis (Fig 4) were between -3.62 and 3.19; -4.57 and 5.07; and 8.27 to 11.6 for the 5-QSI-AP (Fig 4A); the 5-QSI-CS (Fig 4B); and the SDS (Fig 4C) respectively.

## Floor and ceiling effect

For the 5-QSI-CS 19.2% and for the 5-QSI-AP 27.5% of respondents scored the lowest possible score of 0, while the highest possible score of 10 was not recorded for either measure. About 13.3%, 19.2%, and 20.0% of persons with TT scored 1, 2, and 3 on the 5-QSI-AP respectively; while 12.5%, 25.0%, and 7.5% of persons without TT, respectively, scored the same on the 5-QSI-CS. These findings suggest that despite the floor effect, the scales are still capable of distinguishing between those with lower level of stigma scores. The SDS, on the other hand, had a floor effect of 17.5% and no ceiling effect with only 3.3% scoring the maximum value.

**Table 8. Test-retest characteristics of 5-QSI.**

| Tests | Mean (95% CI) | Mean difference (95% CI) | 95% limit of agreement | Intraclass Correlation Coefficient (95% CI) |
|---|---|---|---|---|
| **5-QSI-AP–initial vs repeat** | | | | |
| 5-QSI-AP—initial | 1.82 (1.35–2.28) | 0.22 (-23–0.67) | -3.62 to 3.19 | 0.60 (0.37–0.83) |
| 5-QSI-AP—repeat | 1.60 (1.10–2.09) | | | |
| **5-QSI-CS–initial vs repeat** | | | | |
| 5-QSI-CS—initial | 2.98 (2.36–3.60) | -0.25 (-0.89–0.39) | -4.57 to 5.07 | 0.53 (0.29–0.77) |
| 5-QSI-CS—repeat | 3.23 (2.58–3.89) | | | |
| **SDS–initial vs repeat** | | | | |
| SDS—initial | 7.88 (6.24–9.53) | -1.67 (-2.98 - -0.36) | -8.27 to 11.6 | 0.73 (0.53–0.93) |
| SDS—repeat | 9.55 (7.83–11.27) | | | |

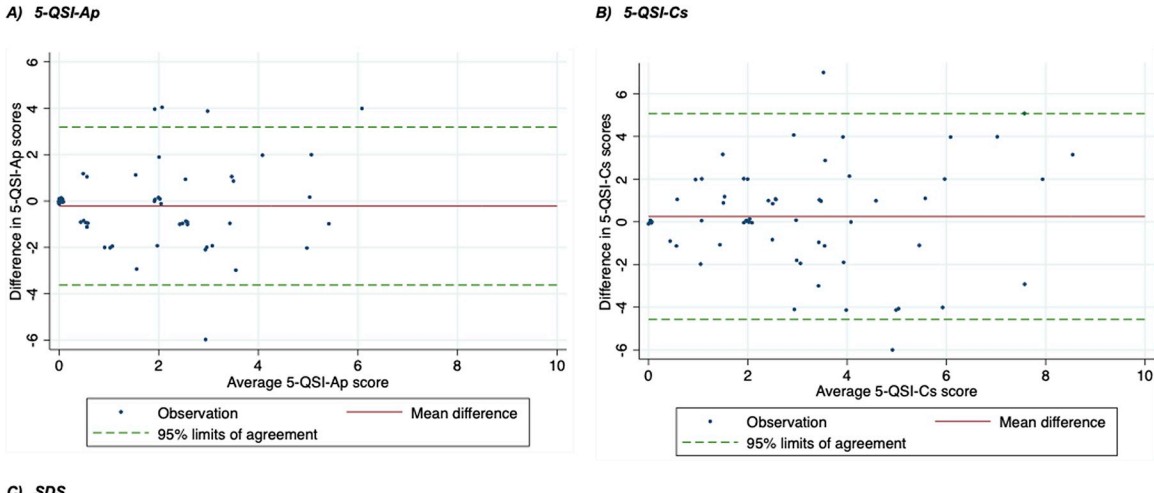

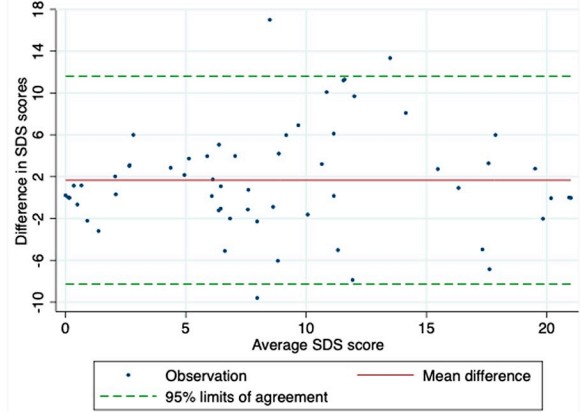

**Fig 4. Bland-Altman plots for test retest of the 5-QSI and the SDS (initial vs repeat assessment).**

A summary of the measurement equivalence results of the 5-QSI and the SDS is presented in Table 9.

## Discussion

The aim of this study was to investigate the cultural validity of the 5-QSI in communities affected by trachoma, mainly the stigma associated with TT. Finding the appropriate term used by the community to express TT was key at the beginning of the conceptual equivalence exercise. We have found it difficult to find a term that adequately describes the blinding sequelae of trachoma. The terms that were suggested in the expert panel were too wordy and technical and often included the term "trachoma" in the translation. However, the most used term by the community doesn't relate trichiasis to "trachoma" and simply translated as "eye hair sprouting" or "eye hair growing." We have also learned that the word "trachoma" is used to express any inflammatory eye disease characterized by inflammation, redness, and discharge. This is probably related to the fact that health promotion messages, which are often delivered in conjunction with mass drug administration campaigns for trachoma, are focused on the active inflammatory stage of the disease that commonly affects children. Therefore, photographs with different severity stages of TT were used to ensure that the study community

**Table 9. A summary of measurement equivalence of the 5-QSI and the SDS.**

| Descriptor | Measurement | Tests Performed | Values | | |
|---|---|---|---|---|---|
| | | | 5-QSI-Ap | 5-QSI-Cs | SDS |
| Participants | | | 120 TT cases | 120 community members | 120 community members |
| Internal consistency | Exploratory Factor Analysis | Kaiser Meyer Olkin test (KMO) | 0.54 | 0.63 | 0.83 |
| | | Bartlett test of sphericity | p<0.001 | p<0.001 | p<0.001 |
| | | Data variance explained by Factor 1 | 87% | 88% | 87% |
| | Reliability coefficient | Cronbach's Alpha | 0.57 | 0.70 | 0.88 |
| Construct validity | Discriminant | Stigma score difference in persons with more severe TT and its sequelae vs those with less severe conditions | With central corneal opacity vs with no or peripheral corneal opacity (mean stigma score 2.48 vs 1.59, p = 0.0084) | - | - |
| | Convergent | Spearman correlation (5-QSI-Cs vs SDS) | - | r = 0.23 | - |
| Repeatability | Test -retest agreement | Limits of agreement from Bland-Altman plots | -3.62 to 3.19 | -4.57 to 5.07 | -8.27 to 11.6 |
| | Test-retest reliability | Intraclass Correlation Coefficient | 0.60 | 0.53 | 0.73 |
| Floor and Ceiling effects | | Floor effect | 27.5% | 19.2% | 17.5% |
| | | Ceiling effect | 0 | 0 | 3.33% |

understood the reference to the blinding stage of the disease, not just all inflammatory eye conditions.

Among the assumptions about trachoma is that it is not associated with stigma. One of the arguments raised in the expert panel discussion to support this assumption was that people with trichiasis often ask people in their villages to epilate them using traditional forceps. However, the qualitative component indicated that stigma and concepts associated with it are linked to TT in this study community. In the FGDs, persons with TT expressed internalized stigma related to the shame of others knowing their condition, indicating that they often "look down" when they meet people, fear people avoid them in relation to their eyesight, and experience insults directed at them and their families. There was also evidence that people affected by TT get stigmatized by the community in different ways, including having problems in existing marriages, particularly for women affected with trachoma; people not wanting to marry them unless they are desperate; or not wanting to share a meal with them. These findings were aligned with a qualitative study conducted in Niger in which people affected with trichiasis indicated the difficulty they face to participate in village activities, the contempt that they received from other people, and people not wanting to eat or have physical contact with them [5].

The relevance and interpretation of item one was questioned in this predominantly agrarian community. Item one asks, "Have you experienced problems in finding or keeping work because you have (had) [condition]" for the 5-QSI-AP and "Would having (had) [condition] cause problems for a person to find work or keep his or her job?" for the 5-QSI-CS. Both persons with and without TT understood that this question refers to difficulty caused by TT experienced in doing their usual work because of the pain or the photophobia from their condition. For instance, one of the FGD participants reported that her neighbor with trichiasis remains asleep to assuage her pain from the trichiasis, and as a result, she faces delays in completing her usual work which should take only a few hours for an unaffected person. The majority of participants involved in the interview were subsistence farmers and are less likely to encounter stigmatization from people around them in relation to their job or occupation.

This forced us to revise the Amharic version of item one to ask if people have tried to not offer them work or not to work with them, which was rather accepted and clearly understood by the study community. Based on the difficulty we faced with item one, we changed the order of the items. The interview started with the shame question (item three) asked first, as it was easily understood by the community, followed by questions on worry others knowing (item four), problems on marriage (item two), people tried to not offer work or not to work with (item one), and people trying to avoid a person (item five) asked at the end as this was another item difficult to understand, as the Amharic term used for "avoid" is also used for "discriminate" in the target culture. We anticipate that similar difficulties may occur in applying this tool in any agrarian community for any disease. We, therefore, recommend that similar revisions be considered on both the source tool and other target culture versions that will be validated in the future.

We used various statistical methods to test the measurement equivalence of the scales. Internal consistency was measured using exploratory factor analysis and reliability coefficient. The 5-QSI-AP had a lower Cronbach's alpha ($\alpha = 0.57$) value than the usually acceptable range and lower than those reported in two studies conducted to assess stigma in leprosy-affected people in Nepal ($\alpha = 0.74$) and ($\alpha = 0.65$) [10, 12]. The difference between the two studies may be related to the difference in the level of stigma toward people affected with leprosy and trachoma. Furthermore, the lower alpha value in this study may not be an indicator of a limitation of the internal consistency of the scale. It is rather likely related to the small number of items (#5) in the tool. It has been previously reported that the chance of finding lower Cronbach's alpha with fewer number of items in a given scale is high and that an $\alpha$ of such size is not uncommon in scales with <10 items [19]. However, the EFA indicated that the 5-QSI-AP is adequately unidimensional with the first factor explaining >80% of the variability in the data, which indicates that the scale is sufficiently internally consistent. Furthermore, the reliability statistics results suggest that the 5-QSI-AP is well-fitted as a 5-item scale. Similarly, the EFA indicated that the 5-QSI-CS is adequately unidimensional with the first factor explaining >80% of the variability in the data. In addition, the Cronbach's alpha of the 5-QSI-CS was in the acceptable range ($\alpha = 0.70$), suggesting strong internal consistency. The $\alpha$ in our study is a bit higher than the results of a study that validated the 5-QSI-CS to monitor leprosy-related stigma in Uttar Pradesh, India ($\alpha = 0.52$) [11]. It is expected that stigma scores are likely to be more internally consistent in a condition like leprosy that results in a more visible physical disability than TT [12]. However, the higher alpha in our study might have occurred from the modification made on item one and the re-ordering of the item administration. The reliability statistics results also suggest that the 5-QSI-CS can be used as a four-item scale without item two.

We measured the construct validity of the 5-QSI in two ways. For the 5-QSI-AP we measured discriminant validity. The discriminant validity results indicated that the 5-QSI-AP measures what it is supposed to measure by showing a mean stigma score difference between those with the advanced stage of the disease and their counterparts. However, the difference in stigma scores between those with major and minor TT and those with and without vision impairment was not statistically significant. This, however, is not unexpected. Firstly, the sample is not powered for such a test. Secondly, as dose-response relationship between stigma and these disease severity parameters is likely to be limited as these parameters are less visible than for corneal opacity or other conditions such as leprosy. Persons with TT with central corneal opacity had a statistically significant higher level of stigma score than those with no or peripheral opacity, suggesting that the tool can pick up the stigma that they will experience if they have a more limiting and/or potentially more visible sequelae of the condition. For the 5-QSI-CS, construct validity was measured using convergent validity with the SDS. This

however showed a weak correlation with the SDS. A study that validated the 5-QSI-CS to monitor leprosy-related stigma and compared it with SDS in Uttar Pradesh, India, reported a moderate correlation (r = 0.50) between the two scales [11]. Both the 5-QSI-CS and SDS are highly likely to provide a stronger correlation in conditions such as leprosy with a clearly determined association with stigma. More importantly, the vignette that precedes the SDS administration may be the reason why such a limited correlation is consistently seen between the SDS and the 5-QSI-CS. It is likely that the SDS paints a rather vivid picture of the condition, which may encourage the interviewee to express their feeling passionately and openly toward people affected by TT.

We measured the repeatability of the scale using agreement and test-retest reliability between the initial and the repeat assessments. The Bland-Altman method did not reveal proportional bias or significant mean difference between baseline and repeat measures, indicating agreement between interviewers for both 5-QSI scales. The 95% limits of agreement could have been more meaningful if it was compared with the minimally important change (MIC). To our knowledge, this is not yet determined for the 5-QSI scales. However, the 95% limits of agreement between the initial and the second assessment stigma scores in our study are narrower than what is reported in a study that validated the 5-QSI-AP in leprosy-affected people in Nepal (-4.79 and 4.83) [10]. We found adequate test-retest reliability for both of the 5-QSI scales, with no evidence of systematic bias between the initial and the repeat assessments. We noticed, however, that the repeat measure for the 5-QSI-CS yielded a higher average stigma score than the initial, which might have led to the wider 95% limit of agreement and relatively lower reliability coefficient. This might have happened due to the learning from the vignette used for the SDS in the initial assessment, which might have in turn biased the responses of the respondents. The SDS score was significantly higher in the repeat than in the initial assessment, suggesting that there was some bias in the way the scale was scored in the repeat visit.

We found a floor effect of >15% and no ceiling effect for both 5-QSI scales. This means that a significant proportion of people had the lowest stigma score (0), while none scored the maximum stigma score (10). Presence of floor effect entails that the people with the lowest stigma scores cannot be effectively distinguished from each other. This is consistent with the findings from other studies conducted on leprosy in Nepal and India for 5-QSI-AP and 5-QSI-CS respectively, which also reported a floor effect of more than 15% and no ceiling effect [11, 12]. Such a higher proportion of "0" or "no stigma" scores however are not unanticipated, since for many stigma scales the normal value is zero. Even more so in our study given that the association of stigma with TT is expected to be modest. Supporting this argument, the stigma scores for TT in both the 5-QSI-AP and 5-QSI-CS were concentrated in the lower half of the distribution, suggesting that when it comes to TT, the 5-QSI doesn't really struggle to differentiate between those with no stigma and those who felt some degree of stigma (scored 1, 2, or 3 on the scale).

## Limitation

This study has several limitations. The revised scale may have less validity in other communities even speaking the same language. We have only validated the 5-QSI as an interviewer-administered scale, as it will be difficult to do so in trachoma-affected communities. Despite the rigorous training that the interviewers took, we cannot rule out the possibility of an influence on the way the participants responded to the questions. The average time taken to administer the 5-QSI was not measured. However, no participant declined to complete the interview during administration, which can be used as a proxy for an acceptably brief tool. The sample size was determined based on the minimum number required for factor analysis. Therefore, it

may not be adequately powered to conduct other tests such as the significance of stigma score differences between subgroups or systematic differences in mean stigma scores between the initial and repeat assessments.

## Conclusions

The findings underpinned the need to conduct a rigorous cross-cultural validation of any tool prior to application from one culture to another. Although some of the psychometric properties didn't show maximum reliability, the 5-QSI-AP and 5-QSI-CS have satisfactory validity to assess internalized stigma in persons with TT and community stigma toward people affected with TT in this Amharic-speaking population. However, this doesn't mean that the Amharic version of this tool should be used uniformly across all Amharic-speaking populations, as the validation process showed that despite the language being the same, some questions can still be understood differently in communities with differing social fabrics (for example dominated by one socioeconomic group). No tool is perfectly valid everywhere. The 5-QSI would benefit greatly from further validation tests in diverse languages and cultures and among people of diverse social backgrounds within the same language or culture. Otherwise, this brief and easy-to-administer scale could offer the possibility to rapidly measure and monitor the stigma associated with TT and other NTDs and health conditions.

## Supporting information

**S1 Data. West Gojam Zone 2022 population data by district and setting.**
(XLSX)

**S1 Table. Social distance scale English version.**
(DOCX)

**S2 Table. Social distance scale Amharic version.**
(DOCX)

**S3 Table. Relevance, and forward and backward translation sufficiency rating guide.**
(DOCX)

## Acknowledgments

We would like to express our deepest gratitude to the persons with TT, community members, and data collectors who participated in this study. We are extremely grateful to experts who hugely contributed to the forward and backward translation, translation sufficiency reviews, and panel discussion: Anna van 't Noordende, Programme Support & Research Officer PEP++, NLR, Amsterdam; Dr. Biset Ayalew, Assistant Professor of English, Bahir Dar University; Dr. Dawit Amogne, Associate Professor of English, Bahir Dar University; Mr. Gebeyehu Mengesha, Associate Professor of Sociology, Bahir Dar University; Dr. Kassahun Habtamu, Assistant Professor of Psychology and Measurement, Addis Ababa University; Dr. Robin van Wijk, Medical Technical Department / Programme Advisor, NLR, Amsterdam; Prof. Takele Azale, Mental Health Epidemiologist, University of Gondar; and Dr. Waltengus Mekonen, Assistant Professor of Folklore, Bahir Dar University.

We would particularly like to mention the generous and unwavering support we received from our funders and the NTD-SC team at The Task Force for Global Health (Elizabeth Long, Emily Hlavaty, and Elias Amin). We would also like to acknowledge our project implementing partners Eyu-Ethiopia, Amhara Regional Health Bureau, London School of Hygiene &

Tropical Medicine, West Gojam Zone Health Department, and the respective district health offices. We are extremely grateful for Laurie Baxley for copy editing the manuscript.

## Author Contributions

**Conceptualization:** Misrak Negash, Zerihun Tadesse, Fentie Ambaw, Michael Beka, Tilahun Belete, Melkamu Abte, Julian Eaton, Eve Byrd, E. Kelly Callahan, David Addiss, Abebaw Fekadu, David Macleod, Matthew Burton, Esmael Habtamu.

**Data curation:** Fentie Ambaw, Esmael Habtamu.

**Formal analysis:** Misrak Negash, Tilahun Belete, Kebede Deribe, David Macleod, Esmael Habtamu.

**Funding acquisition:** Zerihun Tadesse, Fentie Ambaw, E. Kelly Callahan, Matthew Burton, Esmael Habtamu.

**Investigation:** Misrak Negash, Esmael Habtamu.

**Methodology:** Misrak Negash, Zerihun Tadesse, Fentie Ambaw, Michael Beka, Tilahun Belete, Melkamu Abte, Kebede Deribe, Julian Eaton, Eve Byrd, E. Kelly Callahan, David Addiss, Wim H. van Brakel, Abebaw Fekadu, David Macleod, Matthew Burton, Esmael Habtamu.

**Project administration:** Zerihun Tadesse, Esmael Habtamu.

**Supervision:** Zerihun Tadesse, Michael Beka, Tilahun Belete, Melkamu Abte, E. Kelly Callahan, Matthew Burton, Esmael Habtamu.

**Validation:** David Macleod, Esmael Habtamu.

**Writing – original draft:** Misrak Negash.

**Writing – review & editing:** Zerihun Tadesse, Fentie Ambaw, Michael Beka, Tilahun Belete, Melkamu Abte, Kebede Deribe, Julian Eaton, Eve Byrd, E. Kelly Callahan, David Addiss, Wim H. van Brakel, Abebaw Fekadu, David Macleod, Matthew Burton, Esmael Habtamu.

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
