## [Decision Letter · Decision Letter 0]

3 Sep 2024

PMEN-D-24-00313

Cross-cultural adaptation of the 5-question stigma indicators in trachoma-affected communities, Ethiopia

PLOS Mental Health

Dear Dr. Habtamu,

Thank you for submitting your manuscript to PLOS Mental Health. After careful consideration, we feel that it has merit but does not fully meet PLOS Mental Health’s publication criteria as it currently stands. Therefore, we invite you to submit a revised version of the manuscript that addresses the points raised during the review process.

We look forward to receiving your revised manuscript.

Kind regards,

Barkın Köse

Academic Editor

PLOS Mental Health

Journal Requirements:

1. We noticed that you used "unpublished data" in the manuscript. We do not allow these references, as the PLOS data access policy requires that all data be either published with the manuscript or made available in a publicly accessible database. Please amend the supplementary material to include the referenced data or remove the references.

Additional Editor Comments (if provided):

I hope this message finds you well.

Thank you for submitting your manuscript titled "Cross-cultural adaptation of the 5-question stigma indicators in trachoma-affected communities, Ethiopia" to [Journal Name]. The review process for your manuscript has now been completed, and I am writing to inform you of the editorial decision.

Decision: Minor Revision

We kindly ask that you submit your revised manuscript along with a detailed response letter that addresses each of the reviewers' comments. In your response, please indicate how you have addressed each point or provide a rationale if you disagree with a specific suggestion.

Once the revisions have been made, we will promptly reassess your manuscript for final approval. We are confident that these minor adjustments will further strengthen the impact of your research.

Thank you again for choosing Plos Mental Health to publish your work. We look forward to receiving your revised manuscript and hope to proceed with its publication soon.

Best regards,

Reviewers' comments:

Reviewer 1's Comment]

Trachoma, an eye infection caused by Chlamydia trachomatis, is indeed a neglected tropical disease that suffers particularly from stigma. This condition can lead to blindness if not treated properly and is often associated with poor living conditions. Addressing this issue is highly relevant due to its significant impact on affected communities and the need for effective intervention and support.

To improve this article, I recommend considering these observations.

1- To enhance this article, please provide a clear explanation of how you determined the sample size, and ensure this is detailed in the Methods section.

2- It is essential to clearly specify the sampling method adopted, whether it is probabilistic or non-probabilistic.

3- You should develop and include the path diagram, incorporating the correlation coefficients, to provide a clearer representation of the study’s framework.

[Reviewer 2's Comment]

I found this study valuable and believed it would contribute to the literature. Elaborating on just a few points would strengthen the article. My suggestions are as follows:

- The necessity of the study should be detailed in the introduction section, and its purpose and hypothesis(s) should be stated.

- Other scales related to the subject in the literature should be mentioned and the reason for choosing this scale and its advantages over other scales should be explain.

Reviewer's Responses to Questions

**Comments to the Author**

1. Does this manuscript meet PLOS Mental Health’s publication criteria? Is the manuscript technically sound, and do the data support the conclusions? The manuscript must describe methodologically and ethically rigorous research with conclusions that are appropriately drawn based on the data presented.

Reviewer #1: Yes

Reviewer #2: Yes

2. Has the statistical analysis been performed appropriately and rigorously?

Reviewer #1: Yes

Reviewer #2: Yes

3. Have the authors made all data underlying the findings in their manuscript fully available (please refer to the Data Availability Statement at the start of the manuscript PDF file)?

Reviewer #1: Yes

Reviewer #2: Yes

4. Is the manuscript presented in an intelligible fashion and written in standard English?

Reviewer #1: Yes

Reviewer #2: Yes

5. Review Comments to the Author

Reviewer #1: Trachoma, an eye infection caused by Chlamydia trachomatis, is indeed a neglected tropical disease that suffers particularly from stigma. This condition can lead to blindness if not treated properly and is often associated with poor living conditions. Addressing this issue is highly relevant due to its significant impact on affected communities and the need for effective intervention and support.

To improve this article, I recommend considering these observations.

1- To enhance this article, please provide a clear explanation of how you determined the sample size, and ensure this is detailed in the Methods section.

2- It is essential to clearly specify the sampling method adopted, whether it is probabilistic or non-probabilistic.

3- You should develop and include the path diagram, incorporating the correlation coefficients, to provide a clearer representation of the study’s framework.

Reviewer #2: I found this study valuable and believed it would contribute to the literature. Elaborating on just a few points would strengthen the article. My suggestions are as follows:

- The necessity of the study should be detailed in the introduction section, and its purpose and hypothesis(s) should be stated.

- Other scales related to the subject in the literature should be mentioned and the reason for choosing this scale and its advantages over other scales should be explaine

6. PLOS authors have the option to publish the peer review history of their article (what does this mean?). If published, this will include your full peer review and any attached files.

**Do you want your identity to be public for this peer review?** For information about this choice, including consent withdrawal, please see our Privacy Policy.

Reviewer #1: **Yes: **Mohamed Amine BABA

Reviewer #2: No

---

## [Editor Report · Decision Letter 1]

7 Nov 2024

Cross-cultural adaptation of the 5-question stigma indicators in trachoma-affected communities, Ethiopia

PMEN-D-24-00313R1

Dear Dr Habtamu,

We are pleased to inform you that your manuscript 'Cross-cultural adaptation of the 5-question stigma indicators in trachoma-affected communities, Ethiopia' has been provisionally accepted for publication in PLOS Mental Health.

Best regards,

Barkın Köse

Academic Editor

PLOS Mental Health